# TEXT AS PARAMETER: INTERACTIVE PROMPT OPTIMISATION FOR LARGE LANGUAGE MODELS

## ABSTRACT

Large language models (LLMs) can handle a variety of tasks conditioned on natural language instructions. While fine-tuning improves task-specific performance, adjusting the model weights of LLMs requires a huge amount of computational resources, and it is impractical for real-time updates. Alternatively, prompting allows LLMs to adapt to a broad range of tasks without the need for computationally intensive gradient-based optimisation. However, crafting effective prompts remains a challenge, to the extent that it is even unclear if expert in-domain knowledge is what is needed or experience in writing prompts or something else. Approaches like meta-prompting and self-feedback seek to alleviate this burden, but they rely primarily on a numerical feedback signal, leaving the potential of textual feedback unexplored. These methods also typically require numerous interactions with the environment to gather sufficient context, leading to significant computational overhead.

In this work, we propose a novel framework that takes a prompted large language model as an optimiser and treats the text-based prompt itself as a parameter. By interacting with the environment to collect feedback, our proposed method constructs the updated textual prompt. Our experimental results demonstrate that this method not only achieves superior performance but also automatically incorporates domain-specific knowledge, establishing a scientifically motivated, practical and efficient approach to prompting for future research.

## 1 INTRODUCTION

Large language models (LLMs) have shown an extraordinary ability to perform a wide range of tasks, from generating images in various styles to writing code in different programming languages for diverse purposes. These tasks, once considered extremely challenging for artificial intelligence and even beyond the average human's skill set, are now achievable by simply interacting with LLMs. Recently, increasing research efforts have been put into specialisation of LLMs, i.e. improving its capability on specific tasks, in the most effective and efficient manner.

The first family of methods belong to the gradient-based optimisation approaches, such as supervised fine-tuning with low-rank adaptation (Hu et al., 2022), and continuous prompting (Lester et al., 2021; Qin & Eisner, 2021; Li & Liang, 2021; Liu et al., 2023). Although LLMs fine-tuned in this way can excel on the specific target tasks, their in-context learning capabilities may be reduced, and they may also suffer from catastrophic forgetting, losing previously acquired ability when learning the new skill (Zhai et al., 2023; Luo et al., 2023; Wang et al., 2024). In addition, training LLMs via gradient-based optimisation is impractical for real-time updates. The computational requirements for the model size lead to challenges when resources are limited local compute only. More detrimentally, these methods are simply inapplicable to LLMs with only API access.

The second family of methods focus on precise instructions and task descriptions to adapt LLMs via natural language prompting, without updating their model weights. However, creating proper and effective prompts is not straightforward, requiring some tricks, making it at times more similar to art rather than science. Via trial-and-error, various tricks have been found to improve prompting of language models, some of which are widely shared in discussion forums on the internet[1]. Consequently,

---

[1]For example, `https://www.reddit.com/r/ChatGPTPromptGenius/`

approaches like self-feedback or meta-prompting have been introduced to enhance the performance of LLMs without gradient-based optimisation. Self-feedback methods, e.g. LLM-augmenter (Peng et al., 2023) or Reflexion (Shinn et al., 2023), gather sufficient experience by interacting with the environment multiple times and refine the model's response based on verbal summarisation of experiences. In other words, when serving end users, these methods still require numerous queries for self-reflection and refinement of their response, which can cause a huge cost on computation and non-negligible inference time.

On the other hand, meta-prompting methods aim to automatically generate a proper prompt based on a given set of examples or refine a prompt based on scores. Yang et al. (2024) propose Optimisation by PROmpting (OPRO), which employs LLMs to refine an initial prompt based on a number of performance metrics. Tang et al. (2024) introduce the Gradient-inspired LLM-based Prompt Optimizer (GPO), which also uses numerical feedback, controlling edit distance via a cosine decay mechanism. Kong et al. (2024) and Cheng et al. (2024) propose sequence-to-sequence prompt rewriters, trained by reinforcement learning or human preference data, to reduce the difference between human intention and the understanding of LLMs. Although these methods are capable of generating a better prompt, they highly rely on numerical feedback, which is not always available when facing real users.

We propose the *Text-as-Parameter optimisation* (TaP), a meta-prompting approach which leverages natural language textual feedback. Our approach is based on the way humans learn new things: they need instructions, experiences, and feedback. Various studies have shown humans learn better with informative feedback than only simply instruction (Kulhavy, 1977; Mory, 1992; Lyster & Mori, 2006; Hattie & Gan, 2011). To echo this sentiment, in TaP, we first initialise the prompt via corpus-based prompt generation, followed by iterative meta-prompting updates through an interactive optimisation. More specifically, the LLM-based system first interacts with the environment, e.g. real or simulated users trying to complete a particular task such as information seeking or medical question answering. A feedbacker, e.g. human experts or LLMs, provides textual-based feedback based on these interactions, then a separate LLM-based rewriter refines the prompt based on the feedback and the prompt from the previous interaction. The refined prompt is then used by the system on the next user interaction, requiring no additional memory to store previous interactions or past feedback. Textual feedback, which has more information than a scalar or a binary preference, can provide richer information for prompt optimisation.

Our contributions are as follows:

- We propose a Text-as-Parameter optimisation method which leverages textual feedback from external resources for improved prompting.

- Our method can be generalised through different prompting styles, applied with different LLMs, e.g. GPT-4o mini or Gemini-1.5-flash, and is task-agnostic.

- Our method learns domain-specific knowledge through interaction and incorporates it into the optimised prompt. Our experiments show that this reduces the performance gap between different initial prompting methods.

## 2 RELATED WORK

**Gradient-based optimisation for LLMs** For high parameter counts, training or fine-tuning an entire large language model is infeasible since it requires a huge amount of computation resources. As a result, parameter-efficient fine-tuning, such as training only part of the model or freezing the model and training an adapter, is widely used to refine LLMs (Hu et al., 2022; 2023; Lialin et al., 2023). On the other hand, continuous prompting, e.g. prefix-tuning and soft-prompting, is also popular to adapt LLMs to specific tasks or improve their performance (Lester et al., 2021; Qin & Eisner, 2021; Li & Liang, 2021; Liu et al., 2023). By updating inputs of every attention layer (Li & Liang, 2021), or task-related vectors (Lester et al., 2021), these methods can achieve comparable performance to full fine-tuning across various model sizes and tasks (Liu et al., 2022). Although these methods can improve LLMs effectively, they do not apply to API-access-only LLMs and such training processes cannot be carried out in real-time.

**Self-feedback**    To improve the performance of text-based prompts, various prompting styles are proposed, e.g. Chain-of-Thought (Wei et al., 2022) or ReAct (Yao et al., 2023). These prompting methods encourage LLMs to reason before taking action or generating responses, which leads to better performance. However, optimising the prompt for better performance by manual trial and error is inefficient. As a result, self-feedback methods are introduced to refine the LLMs' response, e.g., LLM-augmenter generates feedback by itself and leverages external knowledge to rewrite its response (Peng et al., 2023) and Reflexion summarises previous interactions with the environment as 'reflections' to improve the model's response (Shinn et al., 2023).

While this demonstrates the ability of LLMs for self-correction, these self-feedback methods rely on frequent API calls since their original prompt is not optimal. As a result, the computation cost and latency during inference are not negligible.

**Prompt optimisation**    Meta-prompting methods are widely used to generate a prompt without human editing. The automatic prompt engineer (APE) method leverages an LLM which is instructed to generate an initial prompt and selects the prompt with the best performance on the target task (Zhou et al., 2023). Automatic prompt optimisation (APO) further employs a self-feedback module to provide textual feedback which gives suggestions on how to edit the old prompt (Pryzant et al., 2023). Ye et al. (2024) propose a meta-prompt LLM to edit the original prompt step-by-step. Kong et al. (2024) and Cheng et al. (2024) train a sequence-to-sequence model for prompt rewriting by reinforcement learning and preference data, respectively. Yang et al. (2024) propose optimisation by prompting (OPRO), which leverages LLMs to rewrite the original prompt based on a corresponding performance score. In addition, to properly include experience for improving performance, Zhang et al. (2023) treats LLMs as semi-parametric reinforcement learning agents with a stored experience memory, including task information, observation, action and corresponding $Q$ value estimation. These experiences are sampled dynamically for few-shot in-context learning. Zhang et al. (2024) propose Agent-Pro, which constructs policy-level reflections according to the numerical feedback from the environment and improves its policy incrementally. Tang et al. (2024) introduce the Gradient-inspired LLM-based Prompt Optimizer (GPO), which updates the prompt iteratively based on numerical feedback and controls the edit distance through a cosine-based decay strategy.

Although these methods demonstrate promising performance in generating or improving a prompt, they do not leverage external textual feedback, which could provide abundant information for optimisation. The comparison between our method and the mentioned related works is listed in Table 1.

**Learnability of LLMs with prompting**    Although transformers are universal approximators (Yun et al., 2020) and in-context learning in LLMs can be viewed as implicit fine-tuning (Dai et al., 2023), the following remain open questions: Can we prompt LLMs for arbitrary tasks, and what are the limitations of in-context learning?

Petrov et al. (2024) highlight the limitations of context-based fine-tuning methods, e.g. in-context learning, prompting and prefix tuning, for new task learning in transformers. Specifically, transformers struggle to acquire new tasks solely through prompting, as prompts cannot change the model's attention patterns. Instead, they can only bias the output of the attention layers in a fixed direction and elicit skills learned through pre-training. In other words, only models with billions of parameters trained on vast, diverse datasets are capable of in-context learning—adapting to new tasks through examples or instructions without modifying their underlying weights. Therefore, we focus on fundamental models large enough to demonstrate their in-context learning ability, to investigate text-as-parameter prompt optimisation, which is fully composed of in-context learning with LLMs.

Table 1: Comparison of our proposed method and related works

| Method | What is optimised | Frequent API-calls | Final feedback |
|---|---|---|---|
| Gradient-based | model weights | - | - |
| Self-feedback | - | on inference | self-generated |
| Meta-prompting | system prompt | on optimisation | numerical |
| Text-as-Parameter (ours) | system prompt | on optimisation | external textual |

## 3 TEXT-AS-PARAMETER

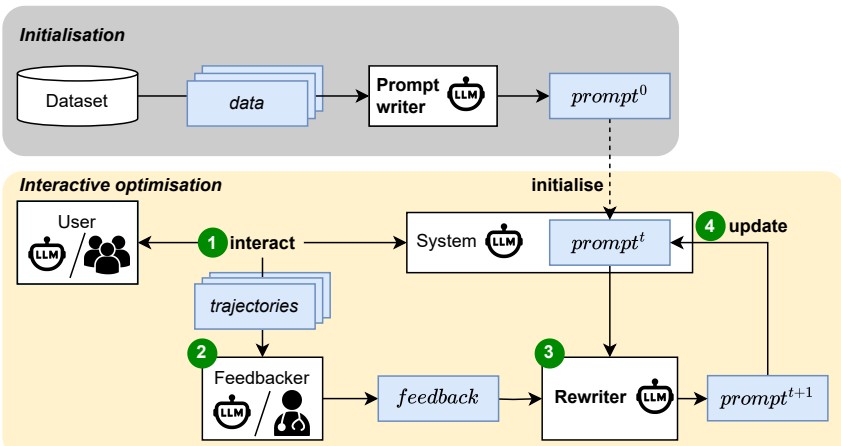

Figure 1: The structure of Text-as-Parameter optimisation. For initialisation, the prompt writer generates an initial $prompt^0$ based on sampled data. In interactive optimisation, the system will first interact with the environment, e.g. simulated or real users. The feedbacker, e.g. human experts or LLMs, will provide textual feedback based on trajectories. The rewriter generates a new prompt based on the original prompt and the textual feedback to update the system's original prompt. One cycle of interactive optimisation is called an epoch.

The structure of the **T**ext-**a**s-**P**arameter (TaP) optimisation method is shown in Figure 1. The initial instruction is generated by a prompt writer $\text{LLM}_P$ in the initialisation phase (the upper part of Figure 1) inspired by the automatic prompt engineer (APE) (Zhou et al., 2023), where we do not provide demonstrations to the prompt writer but APE does. $\text{LLM}_P$ will generate an initial instruction for the system, $prompt^0$, based on a corpus $\mathbb{D}$ for the target task, i.e.

$$prompt^0 \leftarrow \text{LLM}_P(\mathbb{D}'), \tag{1}$$

where $\mathbb{D}'$ is sampled interactions from the dataset to fit into the limited context length of LLMs.

In the interactive optimisation (the lower part of Figure 1), the **system** will interact with the environment, e.g. human users or simulated users, and generate several *trajectories*. These trajectories can be single-turn, e.g. question-answering task, or a multi-turn conversation, e.g. information seeking or recommendation in task-oriented dialogue. Then the **feedbacker**, which can be a language model $\text{LLM}_F$ or human experts, will provide textual feedback to guide the optimisation direction for the **rewriter** $\text{LLM}_R$, which will generate a new prompt to improve the system's performance based on the feedback and original prompt. The improved prompt is then used by the system in the next interaction. That is,

$$feedback = \text{LLM}_F(trajectories), \tag{2}$$

$$prompt^{t+1} \leftarrow \text{LLM}_R(prompt^t, feedback). \tag{3}$$

An example of textual feedback is shown in Figure 2b. It may include strengths, suggestions, and an overall impression for improvement. In contrast, numerical feedback (Yang et al., 2024; Tang et al., 2024) only includes a number quantifying the overall performances, as shown in Figure 2a.

To alleviate human effort for task-specific manual prompt adjustment, our proposed method creates and updates the prompt with LLMs only, where the feedback signal can come from simulated environments or human users. Note that while the feedbacker and rewriter LLMs also require a prompt, they are task-independent and therefore they only need to be engineered once, unlike for the system LLM which may perform a variety of interactive tasks.

The two phases in our method, initialisation and interactive optimisation, are similar to the supervised fine-tuning and reinforcement learning from human feedback respectively. In gradient-based optimisation, the goal is to find the optimal numerical parameters $\theta$, which are initialised by supervised fine-tuning with a loss function $L(\cdot)$ and a dataset $\mathbb{D}$. Then they are further refined through

reinforcement learning using a reward model $f_r(\cdot)$ and an expected return function $J(\cdot)$. In contrast, Text-as-Parameter optimisation seeks to generate the optimal instruction, *prompt*, for an LLM-based system. The prompt is initialised by a prompt writer $\text{LLM}_P$ and is iteratively improved by a rewriter $\text{LLM}_R$ according to feedback from $\text{LLM}_F$. The comparison is summarised in Table 2.

Table 2: Comparison of Gradient-Based Optimisation and Text-as-Parameter Optimisation.

|  | Gradient-based optimisation | Text-as-Parameter optimisation |
|---|---|---|
| Supervised fine-tuning | Learned by gradient descent $\theta^0 \leftarrow \theta - \nabla L(\mathbb{D})$ | Generated by a prompt writer $\text{LLM}_P$ $prompt^0 \leftarrow \text{LLM}_P(\mathbb{D})$ |
| Reinforcement learning | Reward model $f_r$ $r = f_r(trajectories)$ Optimised by policy gradient $\theta^{t+1} \leftarrow \theta^t - \nabla J(\theta^t)$ | Feedbacker $\text{LLM}_F$ $feedback = \text{LLM}_F(trajectories)$ Optimised by Rewriter $\text{LLM}_R$ $prompt^{t+1} \leftarrow \text{LLM}_R(prompt^t, feedback)$ |

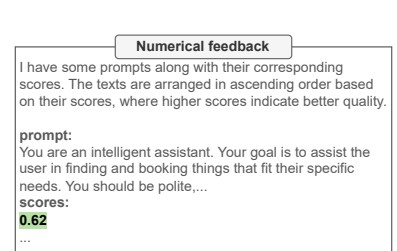

**Textual feedback**

The system did a commendable job assisting the user in planning their trip to Cambridge by providing relevant information about hotel accommodations and train options. Here are some areas of strength and suggestions for improvement:

**Strengths:**
1. **Comprehensive Responses**: The system provided detailed information about hotel options, including amenities like free Wi-Fi and parking, as well as the necessary postcodes.
2. **Train Options**: ...

**Areas for Improvement:**
1. **Direct Booking Function**: ... Incorporating a direct booking capability would enhance user experience by reducing the need for manual follow-up actions.
2. **Inclusion of More Restaurant Options**: ...
3. **Clarification of Pricing**: ...

**Overall Impression:**
The system performed well in addressing the user's needs, ... Implementing the above suggestions could further enhance its effectiveness and user satisfaction for future engagements.

**Numerical feedback**

I have some prompts along with their corresponding scores. The texts are arranged in ascending order based on their scores, where higher scores indicate better quality.

**prompt:**
You are an intelligent assistant. Your goal is to assist the user in finding and booking things that fit their specific needs. You should be polite,...
**scores:**
**0.62**
...

(a) An example of numerical feedback.  (b) An example of textual feedback.

Figure 2: Numerical feedback, used in OPRO (Yang et al., 2024) and GPO (Tang et al., 2024), requires a predefined evaluation function to generate the corresponding score. On the other hand, textual feedback can be generated by LLMs in a simulated environment or by human users.

## 4 EXPERIMENT SETTINGS

In this study, we focus on iterative meta-prompting by leveraging textual feedback from the environment. Our meta-prompting components are task-agnostic (Section 4.1). They are designed to optimise the prompt of the interactive LLM-based systems (Section 4.2). We conduct our experiments on two challenging human-machine interaction tasks, task-oriented dialogue and medical question-answering (Section 4.3). The environment includes simulated users, where the feedbacker is an LLM-based feedbacker or human expert (Section 4.4). More information on optimisation and evaluation can be found in Section 4.5.

Furthermore, to assess how different prompts affect the system's performance, all LLMs in this study are prompted in a zero-shot in-context learning fashion[2], i.e. their prompts solely consist of task descriptions, without any examples or demonstrations included.

### 4.1 META-PROMPTING COMPONENTS

The initial prompt writer $\text{LLM}_P$ in the initialisation phase is built with GPT-4o mini, a cost-efficient model and outperforming GPT-4 (OpenAI et al., 2024) on chat preferences in the LMSYS leaderboard (Zheng et al., 2023). In the interactive optimisation phase, the rewriter $\text{LLM}_R$ is built with

---

[2]Following the definition by Brown et al. (2020), it is in-context learning since the task description is given to LLMs as context, but is also zero-shot because there is no demonstration.

GPT-4o mini or Gemini-1.5-flash (Team et al., 2024). Across different tasks, we keep the prompts of LLM$_P$ and LLM$_R$ fixed, highlighting the task-independent role of these components.

## 4.2 INTERACTIVE SYSTEM

**Task-oriented dialogue**  The dialogue system is built with GPT-4o mini and prompted in two ways, standard and ReAct (Yao et al., 2023). The standard prompt instructs the system to generate responses directly for the user's response (Figure 5a). The ReAct prompt instructs the system to generate its thoughts before responding (Figure 10a). The dialogue system takes the user history as a query and retrieves relevant entities based on Okapi BM25 (Robertson et al., 1995) implemented in langchain[3].

**Medical question-answering**  The system is built with GPT-4o mini and prompted only in the standard way. It responds to users' questions through a single-turn or multiple-turn interaction. In addition, it operates without access to external knowledge bases, relying solely on its knowledge from pre-training for generating answers.

## 4.3 DATASET

**Task-oriented dialogue**  We conduct our experiments on MultiWOZ 2.1 (Budzianowski et al., 2018; Eric et al., 2020), which includes 10k human-to-human conversations related to information-seeking, recommendation and reservation booking over multiple domains. We focus on the attraction, hotel, restaurant, and train domains, i.e., the *database* is composed of entities from these 4 domains and the *requirements* include user goals related to these domains.

**Medical question-answering**  We conduct our experiments on two Chinese medicine datasets, Huatuo-26M (Li et al., 2023) and ShenNong-TCM (Wei Zhu & Wang, 2023). The medical questions in Huatuo-26M and ShenNong-TCM are collected from the internet, e.g. encyclopedias, books, literature, and web corpus, or generated by an LLM based on a traditional Chinese medicine entity graph in Huatuo-26M and ShenNong-TCM, respectively.

## 4.4 USER AND FEEDBACKER

Users are all built with GPT-4o mini for both tasks. The instructions are shown in Figure 8. The simulated users will act based on the user goals, which are descriptions in plain text, such as *"You are looking for a place to stay, the hotel should be in the cheap price range and in the city centre. You also need to find a restaurant nearby."*, or *"我只有咳嗽這一個症狀，請幫我推薦中藥或者方劑。(I only have a cough as a symptom. Please recommend Chinese medicine or a prescription.)"*

While the feedbacker in task-oriented dialogue is built with GPT-4o mini, the feedbacker for the medical question-answering task is human experts, i.e., doctors on general medicine and traditional Chinese medicine. The instructions for the LLM feedbacker are shown in Figure 9.

## 4.5 OPTIMISATION AND EVALUATION

For both tasks, we start by collecting 100 interactions between the user and the system using the initial prompt and goals sampled from the training set. 10 interactions are then sampled randomly and fed to the feedbacker. The sampling is done due to the context length limitation of the LLM-based feedbacker, as well as to leverage the human expert feedbacker wisely. At each epoch, the rewriter generates 5 new prompts based on the previous prompt and feedbacker output. We collect a further 100 interactions for each prompt using user goals sampled from the training set, and the one with the highest score, evaluated by the human expert in medical question-answering or simulated users in task-oriented dialogue, is picked as the new prompt.

**Task-oriented dialogue**  At the end of each turn in the interaction, the user model is instructed to check whether the task is completed, i.e. *"Is the user goal fulfilled? Please check the user goal and the dialogue history carefully. The suggestions from the system must fulfil the user goal, the booked*

---

[3]https://github.com/langchain-ai/langchain

*information should be confirmed, and all requested information should be answered. Please answer 'fulfilled' or 'not finished'."*. The conversation will stop when the goal is fulfilled, i.e. *Complete*, or be forced to quit by the user, i.e. *fail*. For the final evaluation, we collect 100 dialogues with user goals sampled from the test set.

**Medical question-answering** Different systems will interact with simulated patients. Three experts assess these interactions and give their preference based on three factors: safety, professionalism, and fluency. All experts evaluate 30 pairs of interactions on general medicine and 30 on traditional Chinese medicine, where user goals are randomly sampled from the test set.

# 5 RESULTS AND DISCUSSION

## 5.1 TEXTUAL FEEDBACK PROVIDES RICHER INFORMATION FOR OPTIMISATION

Table 3 shows the complete rates of systems optimised by different methods, i.e. without prompt optimisation, optimised with numerical feedback (GPO (Tang et al., 2024)), or our proposed method, optimised with textual feedback (TaP), interacting with an LLM-based user simulator for task-oriented dialogue.

When the rewriter is built with GPT-4o mini, our TaP optimisation pipeline improves the system with a standard prompt by 21% in absolute terms, and the system with a ReAct prompt by 9%. In contrast, the GPO optimisation improves the system with a standard prompt by only 13.1%, with no significant improvement for the system using the ReAct prompt. We observe consistent result on the experiments using Gemini-1.5-flash, as shown in Figure 3. In addition, Figure 3 shows that GPO is barely improved after the first epoch. On the other hand, TaP can still improve the system with more interaction, suggesting a more optimal performance.

These results show that our proposed method is able to improve the model on different prompting styles, standard or ReAct, without limitation between the choice of LLMs, GPT-4o mini or Gemini-1.5-flash. In addition, our proposed method also reduces the gap between the difference between prompting styles, i.e., the performance difference between the standard and ReAct prompt is reduced from 21.6% to 0.32% after optimised by our proposed method with a Gemini-1.5-flash rewrite for 5 epochs. This result suggests our method can provide a more stable, systematic, and scientifically grounded prompt optimisation.

Table 3: Complete rates on 100 MultiWOZ interactions with different optimisation methods. The system is instructed by a standard prompt, i.e. generating responses directly, or by a ReAct prompt (Yao et al., 2023). **F** represents the optimisation algorithm, i.e. without prompt optimisation (-), optimised with numerical feedback (GPO (Tang et al., 2024)), or our proposed method, optimised with textual feedback (TaP). **R** stands for which LLM used by rewriter, i.e. GPT-4o mini (GPT) or Gemini-1.5-flash (Gemini).

| System | F | R | Complete rate | System | F | R | Complete rate |
|---|---|---|---|---|---|---|---|
| Standard | - | - | $0.628_{\pm 0.049}$ | ReAct | - | - | $0.844_{\pm 0.037}$ |
| Standard | GPO | GPT | $0.759_{\pm 0.043}$ | ReAct | GPO | GPT | $0.881_{\pm 0.033}$ |
| Standard | TaP | GPT | $0.846_{\pm 0.036}$ | ReAct | TaP | GPT | $\mathbf{0.934}_{\pm 0.025}$ |
| Standard | GPO | Gemini | $0.829_{\pm 0.038}$ | ReAct | GPO | Gemini | $0.887_{\pm 0.032}$ |
| Standard | TaP | Gemini | $\mathbf{0.974}_{\pm 0.016}$ | ReAct | TaP | Gemini | $\mathbf{0.942}_{\pm 0.024}$ |

## 5.2 WHY THE MODEL'S PERFORMANCE IS IMPROVED THROUGH TAP

**Token counts** Figure 4b shows that no matter which prompting style the system instruction is or which LLM is used for the rewriter, the number of tokens in the prompt is increasing through the optimisation. However, the complete rate is not monotonically increasing, as shown in Figure 4a. This result suggests that simply increasing the length of the prompt does not guarantee an improvement on performance, and that finding specific prompt tokens is important.

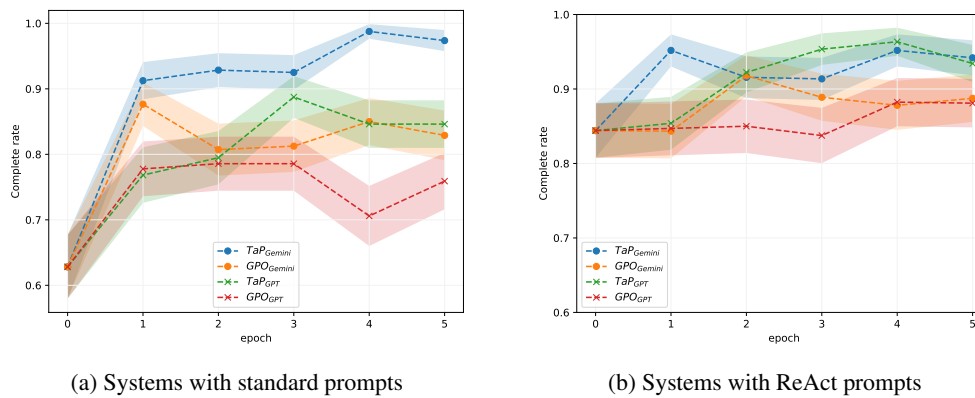

(a) Systems with standard prompts

(b) Systems with ReAct prompts

Figure 3: Result on the Task-oriented dialogue task during optimisation. Systems are updated by GPO and TaP where rewriters are built with different LLMs, i.e. GPT-4o-mini (GPT) or Gemini-1.5-flash (Gemini)

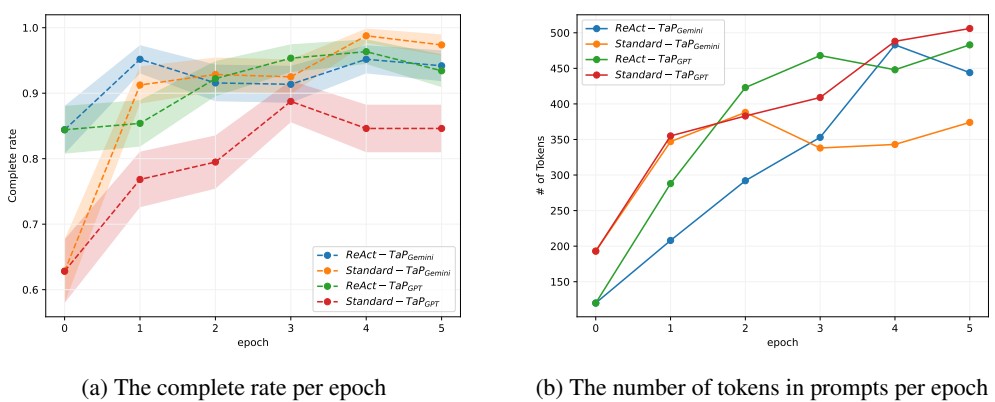

(a) The complete rate per epoch

(b) The number of tokens in prompts per epoch

Figure 4: The rewriter tends to increase the number of tokens in the instructions, but the complete rate is not always improved.

**Domain-specific information is learned through interaction** The initial standard prompt (Figure 5a) only includes vague task definitions such as *"finding and booking things"*, and the initial ReAct prompt (Figure 10a) includes limited task-related information, such as *"book train ticket, hotel room, restaurant reservation, and attraction tickets"* and general booking related slots *dates* and *times*.

Figure 5 and Figure 10 show that the domain-specific information is learnt through interaction. After 5 epochs, more slots related to information seeking, recommendation, and reservation booking are included in the prompt automatically, e.g. *cuisine* and *dietary preferences* for the restaurant domain in Figure 5b, and *number of guests* and *check-in/check-out duration* for the hotel domain in Figure 10b. More booking-related slots, such as *contact information* and *reference numbers* are also included. It is worth mentioning that neither the simulated user, feedbacker nor rewriter have access to any d task-oriented dialogue information since the user goal, dialogues, and system prompts are all presented as plain texts.

According to Table 3, we can find out the system has better performance when more domain-specific information is included in the instruction. This observation is further supported by the fact that human experts also integrate domain-specific knowledge, such as available domains or slots within the context (Heck et al., 2023; Hudeček & Dusek, 2023). In other words, this result suggests our proposed optimisation method can generate a better prompt through interaction because more important domain-specific information is added to the instruction.

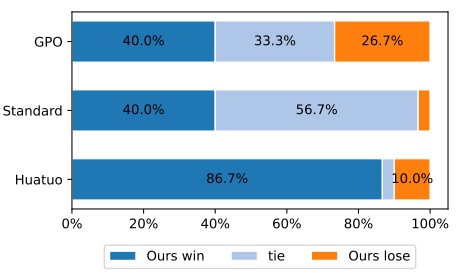

(a) The initial standard prompt.

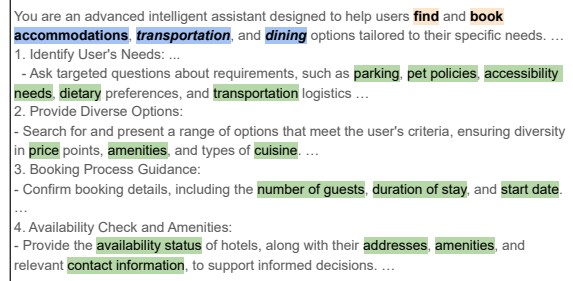

(b) The standard prompt updated by TaP after 5 epochs.

Figure 5: Examples of domain-specific information being learned through interactive prompt optimisation for standard prompts. For illustration, ontology terms, e.g. intent (orange), domain (blue), and slot (green), are manually highlighted based on the ontology in MultiWOZ.

## 5.3 THE LEARNABILITY OF PROMPTING

We compare our method against three systems: a standard system, built with GPT-4o mini with the initial prompt, a standard system updated via GPO, and HuatuoGPT-II (Chen et al., 2024), a large language model which is fully fine-tuned on medical data and demonstrates the state-of-the-art performance on Chinese medicine benchmarks. In other words, except HuatuoGPT-II, a fully fine-tuned 7B model, all systems are built with GPT-4o mini by prompting.

In general medicine, our method consistently outperforms the fully fine-tuned HuatuoGPT-II with an 86.7% win rate and is preferred over other prompting-based baselines. On the other hand, traditional Chinese medicine is more challenging. For example, our system's preference rate drops by 41% compared to Huatuo when transitioning from general medicine to traditional Chinese medicine. However, despite this drop in preference, our proposed method is still favoured in general.

This observation is aligned with the findings by Petrov et al. (2024). Our method performs better in general medicine because the skills present in the pre-training data of LLMs can be elicited by prompting. However, tasks which are unseen or under-represented in pre-training data are hard to learn through prompting. How to properly leverage external knowledge to improve the performance on unseen or under-represented tasks is important in the future.

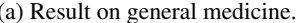

(a) Result on general medicine.

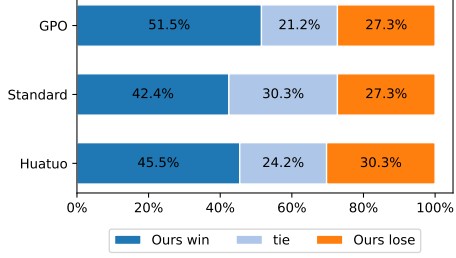

(b) Results on traditional Chinese medicine.

Figure 6: Overall preference between our method and a standard system (Standard), GPO, and HuatuoGPT-II (Huatuo) on the medical question-answering task. The overall recommendation by human experts is based on safety, professionalism and fluency.

## 6 CONCLUSIONS

Our proposed method treats text-based instructions for large language models (LLMs) as parameters. These textual prompts are initialised and optimised by LLMs, without the need for domain-specific manual prompt editing. As shown in Table 3, the method is robust to the choice of LLM used

for rewriting, as it works effectively with both GPT-4o mini and Gemini-1.5-flash. Additionally, it supports various prompting styles, such as standard and ReAct, and reduces the performance gap between these strategies after optimisation over several epochs. By using the optimised prompt, the system can minimise the need for extensive self-feedback loops, reducing computational overhead and API call frequency during inference.

Domain-specific information is learned and encoded in the prompt throughout the interaction with textual feedback (as shown in Figure 5 and Figure 10), enhancing both model performance and explainability. Furthermore, our method can incorporate feedback from simulated environments as well as human experts, which enables a more feasible way to incorporate human expert knowledge. Our method offers a stable, practical, and efficient approach for automatic prompt optimisation, which could be valuable for future LLM research.

### ETHIC STATEMENT

This work uses open-source datasets, such as MultiWOZ, Huatuo-26M, and ShenNong-TCM. The MultiWOZ dataset is widely used in research on task-oriented dialogue. The Huatuo-26M dataset is collected from publicly accessible data without personal information and is available to academic researchers. The ShenNong-TCM dataset is generated by GPT-3.5 based on a traditional Chinese medicine knowledge graph. As a result, these datasets should not be regarded as controversial. All interactions are generated by LLMs, which may inevitably include hallucinations or in-correct information. Human evaluators are also fully aware they are reading interactions generated by LLMs.

### REPRODUCIBILITY STATEMENT

The datasets used in this work are all open-sourced. The `GPT-4o mini` model is available through OpenAI's platform `https://openai.com/api/`. The `Gemini-1.5-flash` model is available through Google Cloud Platform Vertex AI. In this work, we use the versions of the model with identifiers `gpt-4o-mini-2024-07-18` and `Gemini-1.5-flash-001`.

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

## A PROMPTS FOR LLMS IN TAP

Here are the prompts for components in our proposed method.

---

You are an assistant tasked with improving the prompt instruction of another large language model assistant.
You will be given the previous instruction prompt and its feedback.
Please generate a new instruction prompt for the next iteration, **with performance improvement**.
Please output the new instruction prompt directly without any extra description, since the result would be fed back into the assistant directly. The new prompt should not be longer than 512 tokens.

Here is the previous instruction `[OLD PROMPT]` and the feedback `[FEEDBACK]`

---

Figure 7: The prompt for the rewriter.

---

You are a user and try to use a task-oriented dialogue system to accomplish your goal. Here are the guidelines for the conversation:

**1. Provide Information:**
Answer the assistant's questions clearly and provide all necessary information.

**2. Ask for Clarification:**
If you are unsure about something, ask the assistant to clarify.

**3. Confirm Booking:**
Check the booking details provided by the assistant and confirm if they are correct and align with the user goal.

**4. End the Conversation:**
Say 'goodbye' after the system fulfils your user goal, e.g. providing all information in the user goal, answering all your requests, or making a reservation correctly.

If you are unsatisfied, e.g. the system keeps providing recommendations which do not fulfil your goal or the system is not helpful, you should say 'quit' at any time.

Here is the user goal: `[USER GOAL]`

---

Figure 8: The prompt for user simulators in task-oriented dialogue systems
.

Based on the user goal and the dialog history, please provide feedback to the system. The feedback should be constructive and helpful for the system to improve.

Here are the user goals `[USER GOALS]` and the dialogs `[DIALOG]`

Figure 9: The prompt for feedbacker in task-oriented dialogue systems
.

You are an intelligent assistant tasked with helping users **book** *train* tickets, *hotel* rooms, *restaurant* reservations, and *attraction* tickets, as well as providing information ...
1. Break down the user's request into specific subtasks.
2. Reason through each step before taking any action.
3. Ask the user for necessary information to complete the task, such as dates, times, preferences, or other relevant details.
4. Provide available options ...
5. Respond clearly, concisely, and politely.
Please generate your reasoning first, starting with `[Think]`, and then your response, starting with `[Response]`.

(a) The initial ReAct prompt.

You are an intelligent assistant designed to help users **book** *train* tickets, *hotel* rooms, *restaurant* reservations, and *attraction* entries. ...
1. Strictly Follow User Specifications: ...
2. Decompose Requests: ...
3. Clarify User Preferences: Actively prompt users to share detailed preferences, such as cuisine, ambience, and specific needs, to provide personalized recommendations.
4. Gather Comprehensive Information: Collect all necessary details for bookings, including travel dates, number of guests, and specific requirements. Suggest standard options like typical check-in/check-out durations to facilitate quicker decisions.
5. Present Varied Choices: Offer a range of relevant options, summarizing key details like pricing and location. ....
7. Deliver Detailed Confirmations: Provide thorough confirmations for all bookings, including dates, times, reference numbers, ....
Please generate your reasoning first, starting with `[Think]`, and then your response, starting with `[Response]`.

(b) The ReAct prompt updated by TaP after 5 epochs.

Figure 10: The domain-specific information is learned through the interactive prompt optimisation for ReAct prompts. The ontology terms, e.g. intent (orange), domain (blue), and slot (green), are manually highlighted based on the ontology in MultiWOZ.

