# OpenReview forum: "Text as parameter: interactive prompt optimisation for large language models"
_ICLR.cc/2025/Conference — ICLR 2025 Conference Withdrawn Submission_

### Official Review · Reviewer_P5iy · 2024-11-01

**Soundness:** 2
**Presentation:** 2
**Contribution:** 1
**Rating:** 3
**Confidence:** 4

**Summary:**

This paper proposes a method for prompt optimization that utilizes text feedback from an optimizer large language model (LLM). The proposed method is compared with GPO, a method that optimizes prompts with numerical feedback (such as accuracy score). The proposed TaP method significantly outperforms the GPO baseline by a notable margin on 1000 MultiWOZ and on two Chinese medicine datasets.

**Strengths:**

- This paper presents an excellent comparison between prompt optimization and other methods like prompt tuning and inference-time self-refinement.
- Experiments on two challenging human-machine interaction tasks demonstrate that this method not only achieves superior performance but also automatically incorporates domain-specific knowledge

**Weaknesses:**

- The paper is lacking in comprehensive comparisons with recent baselines. For example, the related work introduced between lines 120 to 135, like APO and OPRO. However, the experiment only compares with GPO, which makes the empirical result rather weak.
- The novelty of the paper is also lacking. As the author pointed out in line 120, the APO method also uses "textual feedback which gives suggestions on how to edit the old prompt". Having read the APO paper, it is unclear to me how the proposed method differs from APO, except for the difference in meta prompts.

**Questions:**

- How is your method differnt from APO?

---

> ### Author Response · Authors · 2024-11-27
>
> The internal feedback (textual gradient $g$) in APO [1] depends on the input and output pairs $e = [(x_i, y_i): (x_i, y_i) \in D]$ and the original prompt $p$, i.e. $[g_1, \dots, g_m]=LLM_\nabla(p, e)$, on the other hand, the external feedback in our method depends on the input and output pairs generated from interaction only, i.e. $feedback = LLM(e)$. In this way, we can leverage external feedback without revealing the original prompt. In addition, our proposed method can also improve over different initial prompting strategies and leverage feedback from human experts in the medical Q\&A.
>
> It is also worth mentioning that GPO, our major baseline, has reported that GPO outperformed APO and OPRO in various tasks.
>
> [1] Automatic Prompt Optimization with “Gradient Descent”
> and Beam Search, EMNLP 2023

---

### Official Review · Reviewer_TnP7 · 2024-11-04

**Soundness:** 2
**Presentation:** 3
**Contribution:** 2
**Rating:** 5
**Confidence:** 4

**Summary:**

This paper introduces a novel framework called Text-as-Parameter (TaP) for optimizing prompts in LLMs based on the way humans learn new things. Instead of relying on fine-tuning or simple numeric feedback, TaP treats the prompt text itself as a parameter, iteratively refining it based on textual feedback from interactions. The process involves initializing a prompt, then continually updating it through a feedback loop where a ``rewriter'' component incorporates the feedback to generate an improved prompt for the next interaction. Experimental results indicate that TaP outperforms existing numerical-feedback methods in diverse applications, including task-oriented dialogue and medical question answering.

**Strengths:**

* Using detailed textual feedback instead of simple numerical scores is intuitively beneficial, as it provides richer, more nuanced guidance for prompt adjustments.

* The paper presents a comprehensive set of experiments demonstrating the benefits of TaP across simulated and real-world settings, thus validating the approach’s effectiveness compared to traditional numerical feedback methods.

* The TaP method demonstrates steady improvement in complete rates over multiple epochs, indicating its robustness and potential for long-term usability.

**Weaknesses:**

* While the shift from numerical scores to textual feedback enhances prompt refinement, this alone may not constitute a sufficient contribution.

* The claim of treating text as a “parameter” feels overstated. While the method refines prompts iteratively, it primarily mirrors traditional prompt optimization techniques and doesn’t fully leverage text as an integrated parameter of the model.

* Table 2 presents a comparison by aligning TaP optimization closely with gradient-based methods. However, unlike gradient-based methods which iteratively refine continuous parameters towards optimal values, TaP relies on discrete prompt rewriting. This could lead to inconsistent improvements, as prompt optimization depends heavily on the quality of feedback and may not consistently yield better outcomes.

**Questions:**

* When both the user, rewriter, and system are built using LLMs, the framework essentially functions as an LLM-based multi-agent system. Given this structure, it would be valuable for the authors to explore comparisons with existing multi-agent collaboration methods. Have the authors considered benchmarking TaP against multi-agent collaboration methods?

[1] Kim D K, Sohn S, Logeswaran L, et al. MultiPrompter: Cooperative Prompt Optimization with Multi-Agent Reinforcement Learning[J]. arXiv preprint arXiv:2310.16730, 2023.

---

> ### Author Response · Authors · 2024-11-27
>
> Our method demonstrates steady improvement across different tasks, initialised prompting style, and LLMs. It provides a sufficient way to incorporate domain-specific knowledge from human experts who may not be familiar with prompting strategies or large language models.
>
> Thanks for your suggestion, we will include a comparison with multi-agent collaboration methods.

---

### Official Review · Reviewer_wSpN · 2024-11-04

**Soundness:** 2
**Presentation:** 3
**Contribution:** 2
**Rating:** 3
**Confidence:** 4

**Summary:**

This paper introduces a novel framework for optimizing prompts in large language models (LLMs) by treating text-based prompts as parameters that can be iteratively improved through feedback interactions. The proposed method leverages textual feedback, refining prompts based on interactions with the environment, which integrates domain-specific knowledge. Experimental results show that the method is effective across various LLMs, such as GPT-4o mini and Gemini-1.5-flash, and works well with multiple prompting styles, including standard and ReAct.

**Strengths:**

(1) The framework proposed in this paper is effective. Without additional training or fine-tuning, TaP improves performance across prompting styles under the GPT4-o and Gemini.

(2) The framework diagram in the paper is well-crafted.

**Weaknesses:**

(1) The evaluation dataset is small and lacks benchmark comparisons on popular datasets. The experiments in this paper are conducted on 100 MultiWoZ instances, 30 pairs of interactions in general medicine, and 30 in traditional Chinese medicine. In contrast, the baseline method GPO [1] provides comparison results across multiple datasets (BBH, GSM8K, MMLU, WSC, WebNLG). Extending this method to more widely used evaluation datasets would enhance its reliability and effectiveness.

(2) The evaluated models are limited and are all closed-source, namely GPT-4o and Gemini. It remains to be seen whether this approach applies to different sizes of open-source models. I suggested extending the method to the Llama-2 series, as GPO does, to enable a direct performance comparison between your method and GPO.

(3) The method lacks novelty; the instruction-feedback-refine framework is familiar in NLP.

[1] Unleashing the Potential of Large Language Models as Prompt Optimizers:
An Analogical Analysis with Gradient-based Model Optimizers

**Questions:**

1. Why did you choose to evaluate Task-oriented dialogue and medical question-answering tasks rather than using popular benchmark datasets?

2. Why was the evaluation conducted on small-scale samples (e.g., 30 samples) from three datasets instead of using the entire test set? Evaluating just a few selected samples seems like cherry-picking, which may lead readers to question the model's effectiveness.

---

> ### Author Response · Authors · 2024-11-27
>
> **The choice of benchmarks**
>
> We conducted experiments on tasks hard to evaluate numerically, e.g. user satisfaction in task-oriented dialogues and safety in medical Q\&A can be represented more properly by textual feedback.
> The testing samples are selected randomly, but we will conduct larger-scale experiments.
>
> **Comparison with the instruction-feedback-refine framework**
>
> The self-refine (or self-feedback) methods are different to our method since they modify the generated response instead of optimising the prompt (as shown in Table 1), where these methods require frequent API calls during inference.

---

### Official Review · Reviewer_yUvK · 2024-11-04

**Soundness:** 1
**Presentation:** 2
**Contribution:** 2
**Rating:** 3
**Confidence:** 3

**Summary:**

In this paper, the authors propose a prompt optimization method called Text-as-Parameter. In this framework, the initial prompt is used to generate some samples which consist of interaction between LLM and users. Then, the interactions are sent to feedback LLM to generate a review and then sent to rewrite to rewrite the prompt. The results on two datasets show that the proposed TaP outperforms numerical feedback.

**Strengths:**

1. Prompt optimization is an important and urgent topic for LLMs as prompt influences the performance significantly and it is still unknown how to find the best or robust prompt.

2. The idea of self-improvement or refinement is popular and seems promising.

3. The experimental results on MultiWOZ improve the numerical baseline by a large margin.

**Weaknesses:**

1. The experiments are not solid. Although the introduction and related work mention a number of works such as APO, OPPO, and GPO. The experiments only compare with one GPO baseline, weakening the conclusions. Besides, more ablation studies are needed to understand how the proposed framework works. Also, it is unclear how the GPT-simulated user performs. At least some human annotations are needed to confirm the simulation quality. Overall, it is difficult to judge the performance of the entire system.

2. The novelty of the proposed work is limited. If I understand correctly, the biggest novelty is the external text feedback rather than the numerical ones. However, there are some studies generating text-based feedback for optimizing prompts, such as TextGrad [1]. Also, textual feedback and rewriting are well-established, such as self-refine [2,3]. This further weakens the novelty of the proposed work.

3. Writing and presentation need to improve. For instance, the introduction does not introduce the experimental results.  And more explaination is needed for trajectories and other designs rather than proposing a name alone.

[1] Yuksekgonul et. al. TextGrad: Automatic "Differentiation" via Text

[2] Madaan et. al. SELF-REFINE: Iterative Refinement with Self-Feedback

[3] Wadhwa et. al. Learning to Refine with Fine-Grained Natural Language Feedback

**Questions:**

See above weakness.

---

> ### Author Response · Authors · 2024-11-27
>
> **Ablation Study**
>
> In Table 3, we show the results of all combinations over different initialised prompting methods (Standard or ReAct), different optimisation methods (no optimisation, optimised by numerical feedback, i.e. GPO, and optimised by textual feedback, i.e. our proposed method TaP), and different LLMs for rewriter (GPT-4o-mini or Gemini-flash).
> It shows that our proposed method can be generalised across different LLMs, bridge the gap between different initial prompting strategies, and outperform the system optimised by numerical feedback.
>
> It is also worth mentioning that GPO, our major baseline, has reported that GPO outperformed APO and OPRO in various tasks.
>
> **Comparison with other methods**
>
> * In comparison with TextGrad [1], our method leverages external feedback signals, i.e. the feedbacker does not access the original prompt, in addition, leveraging with human feedback is not discussed in their technical report or GitHub repo.
> * The self-refine (or self-feedback) methods [2,3] are different to our method since they modify the generated response instead of optimising the prompt. As shown in Table 1, these methods require frequent API calls during inference.
>
>
> Thanks for your advice, we will improve our presentation in our next version and include an analysis of the GPT-based user simulator and results of other baselines such as APO and OPRO.
>
> [1] TextGrad: Automatic “Differentiation” via Text, ArXiv 2024
>
> [2] SELF-REFINE: Iterative Refinement with Self-Feedback, NeurIPS 2023
>
> [3] Learning to Refine with Fine-Grained Natural Language Feedback, EMNLP 2024

---

### Official Review · Reviewer_6mtw · 2024-11-06

**Soundness:** 3
**Presentation:** 2
**Contribution:** 2
**Rating:** 3
**Confidence:** 4

**Summary:**

The paper proposes Text-as-Parameter (TaP) to optimize prompts. This method involves interacting the model with the environment and using another model to provide detailed textual feedback that discusses the strengths and limitations of the prompt. The prompt is then rewritten based on this feedback. Experiments demonstrate that this approach achieves good performance in task-oriented dialogue and medical question-answering domains.

**Strengths:**

1. The paper proposes Text-as-Parameter (TaP), a method that leverages textual signals as feedback to iteratively optimize prompts.
2. Experiments on task-oriented dialogue and medical question-answering demonstrate the effectiveness of the method.

**Weaknesses:**

1. The novelty of this work lies primarily in replacing the score-based evaluation with a text-based evaluation to measure the prompt quality. But this is just one aspect of various kinds of feedback in previous works[1][2][3]. I believe the authors have overstated the contribution of the paper.

2. The authors only verify their methods on closed LLMs and do not evaluate open-source LLMs, such as Llama-3 and Qwen-2. They also miss to compare their approach with some recent baselines, such as [1][2]. Additionally, they fail to assess their methods on general benchmarks, such as BBH and MMLU, which are commonly used by other baselines.

3.The authors do not provide a detailed analysis of some important characteristics of the method, such as convergence, generalization, and the impact of the initial prompt in prompt optimization.

[1] PromptAgent: Strategic Planning with Language Models Enables Expert-level Prompt Optimization, ICLR 2024

[2] Prompt engineering a prompt engineer, ACL 2024

[3] Unleashing the Potential of Large Language Models as Prompt Optimizers: An Analogical Analysis with Gradient-based Model Optimizers, arXiv 2024

**Questions:**

See the weaknesses part.

---

> ### Author Response · Authors · 2024-11-27
>
> **Comparison with other methods**
>
> Our method aims to ease the effort of manually prompt engineering, where the initial prompt is generated automatically based on the dataset and keeps improving iteratively based on feedback from LLMs or human experts.
>
> * In PromptAgent [1], the initial prompt is human-written, on the other hand, our prompt optimisation is fully machine-generated without human-written task-specific prompts.
> * In PE2 [2], the same LLM generates the feedbacker and new prompt.
>     Still, our method can leverage external feedback, purely based on the system's behaviour without accessing the original prompt.
>     In addition, our method can optimise over multiple epochs but PE2 can barely improve after 2 epochs.
> * The GPO [3] leverages the numerical feedback and we show significant improvement in our experiment results.
>
> Furthermore, these methods [1,2,3] did not test with various initial prompt styles, e.g. a standard method or ReAct prompting method, on the other hand, our method can bridge the difference between various prompting styles.
> We will clarify the difference between our method and the other works in our next version.
>
> **The choice of benchmarks and baselines**
>
> We aim to conduct experiments on tasks difficult to measure only by numerical metrics, e.g. user satisfaction in task-oriented dialogues and safety in medical Q\&A can be represented more properly by textual feedback.
> It is also worth mentioning that GPO, our major baseline, has reported that GPO outperformed PE2 in various tasks.
>
> **Analysis of generalisability and the impact of initial prompts**
>
> We show our method can be generalised across two different tasks (task-oriented dialogue and medical Q\&A), different languages (English in task-oriented dialogue and Chinese for medical Q\&A), different LLMs (GPT-4o-mini and Gemini-flash), and different feedback source (LLM or human expert). In addition, we tested with different initialised prompting methods (standard or ReAct) and our proposed method can improved over all these settings.
>
> [1] PromptAgent: Strategic Planning with Language Models Enables Expert-level Prompt Optimization, ICLR 2024
>
> [2] Prompt engineering a prompt engineer, ACL 2024
>
> [3] Unleashing the Potential of Large Language Models as Prompt Optimizers: An Analogical Analysis with Gradient-based Model Optimizers, arXiv 2024

---

### Author Response · Authors · 2024-11-27
**General response**

Thank you for taking the time to read our manuscript and provide valuable comments. Note that our method mainly targets the LLMs available via an API, i.e., those that cannot be manipulated using fine-tuning and where not even logits are available. However, we can add the open-access LLMs for comparison in the next iteration.

---

### Note · Authors · 2024-12-12

I have read and agree with the venue's withdrawal policy on behalf of myself and my co-authors.